# Evaluation of reporting quality of randomized controlled trials in patients with COVID-19 using the CONSORT statement

Yuhuan Yin[1‡], Fugui Shi[2‡], Yiyin Zhang[1], Xiaoli Zhang[1], Jianying Ye[1], Juxia Zhang[3]*

**1** School of Nursing, Gansu University of Chinese Medicine, Lanzhou, China, **2** Lanzhou Hand and Foot Surgery Hospital, Lanzhou, Gansu, China, **3** Clinical Educational Department, Gansu Provincial Hospital, Lanzhou, Gansu, China

‡ These authors share first authorship on this work.
* 1425331412@qq.com

**Data Availability Statement:** All relevant data are within the manuscript and its Supporting Information files.

## Abstract

### Objective

To evaluate the reporting quality of randomized controlled trials (RCTs) regarding patients with COVID-19 and analyse the influence factors.

### Methods

PubMed, Embase, Web of Science and the Cochrane Library databases were searched to collect RCTs regarding patients with COVID-19. The retrieval time was from the inception to December 1, 2020. The CONSORT 2010 statement was used to evaluate the overall reporting quality of these RCTs.

### Results

53 RCTs were included. The study showed that the average reporting rate for 37 items in CONSORT checklist was 53.85% with mean overall adherence score of 13.02±3.546 (ranged: 7 to 22). The multivariate linear regression analysis showed the overall adherence score to the CONSORT guideline was associated with journal impact factor (P = 0.006), and endorsement of CONSORT statement (P = 0.014).

### Conclusion

Although many RCTs of COVID-19 have been published in different journals, the overall reporting quality of these articles was suboptimal, it can not provide valid evidence for clinical decision-making and systematic reviews. Therefore, more journals should endorse the CONSORT statement, authors should strictly follow the relevant provisions of the CONSORT guideline when reporting articles. Future RCTs should particularly focus on improvement of detailed reporting in allocation concealment, blinding and estimation of sample size.

**Funding:** The study was funded by Horizontal project of School of Public Health of Lanzhou University "Research on the Thinking, difficulties and Countermeasures of High-quality Development of Lanzhou Hand and Foot Surgery Hospital" (071100278), Gansu Science and Technology Plan (Innovation Base and Talent Plan) project (21JR7RA607), and Health industry scientific research project of Gansu Province (GSWSKY-2019-50), received by Fugui Shi and Juxia Zhang.

**Competing interests:** The authors have declared that no competing interests exist.

## Introduction

With the development of evidence-based medicine, randomized controlled trials (RCTs) became the gold standard to compare the effectiveness of different interventions [1]. It can avoid possible bias in clinical trial design, balance confounding factors and improve the effectiveness of statistical tests [2]. Therefore, a complete and accurate report will enable readers to fully assess the authenticity of results [2]. If the RCT report is unsatisfactory, the validity of trials will be reduced [3], which may adversely affect the results of meta-analyses and recommendations for clinical practice [4].

To improve the reporting quality of RCTs, the CONSORT (reporting standards of randomized controlled trials) statement was developed in 1996 [5], revised in 2001 [6] and 2010 [7]. The updated CONSORT statement includes 25 entries that provide specific guidance for RCT reports. Since many of the 25 items were subdivided into two subitems, the list actually consists of 37 items [7]. It is currently known to be endorsed by over 600 biomedical journals and endorsed by several prominent editorial organizations including the International Committee of Medical Journal Editors (ICMJE) and the World Association of Medical Editors (WAME) [8]. After update of CONSORT statement, the overall reporting quality of RCTs has improved, but there were still many RCT reports with various deficiencies [9, 10]. A study by Yao et al. of 65 RCTs related to ophthalmic surgery showed the mean CONSORT score was 8.9 (range 3.0–14.7) and the reporting quality was quite low [11]. A study of 71 RCTs regarding herbal interventions showed that the compliance rates of CONSORT checklist in these RCTs ranged from 0% to 97.18% [12]. Poor reporting quality of RCTs is the major barrier to evidence-based practices, as it can distort the available evidences in the medical literature, and prevent clinical decision-makers from obtaining true results from trials [13]. Thus, it is critical for researchers to build well-reported standard of RCTs. On the one hand, it can ensure the validity of clinical trials and the authenticity and scientificity of research results [14]. On the other hand, it is conducive to conducting secondary studies, such as meta-analyses and systematic reviews [15].

With the outbreak of the COVID-19 in 2019, in order to quickly and effectively control the epidemic, a large number of RCTs regarding COVID-19 have been published. However, the reporting quality of these RCTs is unclear, to our knowledge, no study has specifically evaluated the reporting quality of RCTs regarding patients with COVID-19.

The primary objective of our study was to assess the reporting of RCTs regarding patients with COVID-19 and analyze possible related factors, so as to provide theoretical basis for subsequent studies and meta-analyses.

## Methods

### Ethical review

Ethical approval was not necessary for this study, as the study did not involve patients and included RCTs can be traced from databases.

### Search strategy

PubMed, Embase, Web of Science and the Cochrane Library databases were searched to collect RCTs regarding patients with COVID-19. The retrieval time was from the inception to December 1, 2020. The search was conducted by two investigators and the detailed strategy was shown in S1 File.

## Study selection

Studies meeting the following criteria were enrolled in the study: (1) Randomized controlled trial. (2) The confirmed or suspected patients of COVID-19 according to the diagnostic criteria of "the latest Clinical guidelines for novel coronavirus" issued by the World Health Organization (WHO). (3) Interventions related to patients or suspected patients.

Exclusion criteria included: (1) Animal experiments, reviews, systematic reviews, case reports. (2) Repeated publications. (3) The abstract or full text is not available.

The titles of the retrieved article were imported into the Endnote X9 and screened by 2 reviewers independently. We first reviewed the title and abstract of each article and decided to regard its appropriateness for inclusion. In case of doubt, we downloaded full texts to judge whether an article was RCT. Any disagreement was solved by consensus.

## Data extraction

Two authors independently extracted the general characteristics and reporting data of included studies into Excel, any discrepancy was resolved through discussion. The general characteristics include continent of first-author, number of authors, sample size, age and type of participants, the type of interventions, journal impact factor and journals' endorsement of the CONSORT statement.

## Assessment of reporting quality

The CONSORT statement was chosen as a tool to assess reporting quality of these RCTs [7]. We assessed the compliance of each RCT by 25 items of CONSORT statement, each checklist item and subitem was answered with "yes" or "no". According to the above items, the coincidence rate of each item of 53 studies was counted one by one. A point for the item being granted if all sub-sections were answered as yes, if one of two subsections was reported, a score of 0.5 was awarded, then total score of each study was calculated [1]. Items 3b, 6b, and 14b are not necessarily applicable, if relevant, the article will be graded according to the above guidelines, if the article did not apply to this item, no points was deducted [3].

## Statistical analysis

Descriptive statistics was performed to describe general characteristics of 53 studies and the reporting rate of each checklist item/subitem. We used the k coefficient to determine the degree of agreement between reviewers. T-test and ANOVA were used for univariate analysis, multiple linear regression analyses were used to determine the association between potential predictors and reporting quality. All significant predictors in the univariable analyses were entered individually into a multivariable analysis. No significant violation of normality was found in assessments of the residuals. Chi-square tests and Fisher's exact tests were used to analyze the relationship between journals impact factor and endorsement of CONSORT guideline, T-test was used to analyze the differences between different journal submission requirements and CONSORT score. For all analyses, the statistical significance level was set at P<0.05. Statistical analysis was carried out using SPSS version 21.

# Results

## Search results

Initially, 8700 articles were obtained, excluding duplicates, 6,922 studies was remained. After screening the titles and abstracts, 198 potentially eligible articles were identified. Subsequently

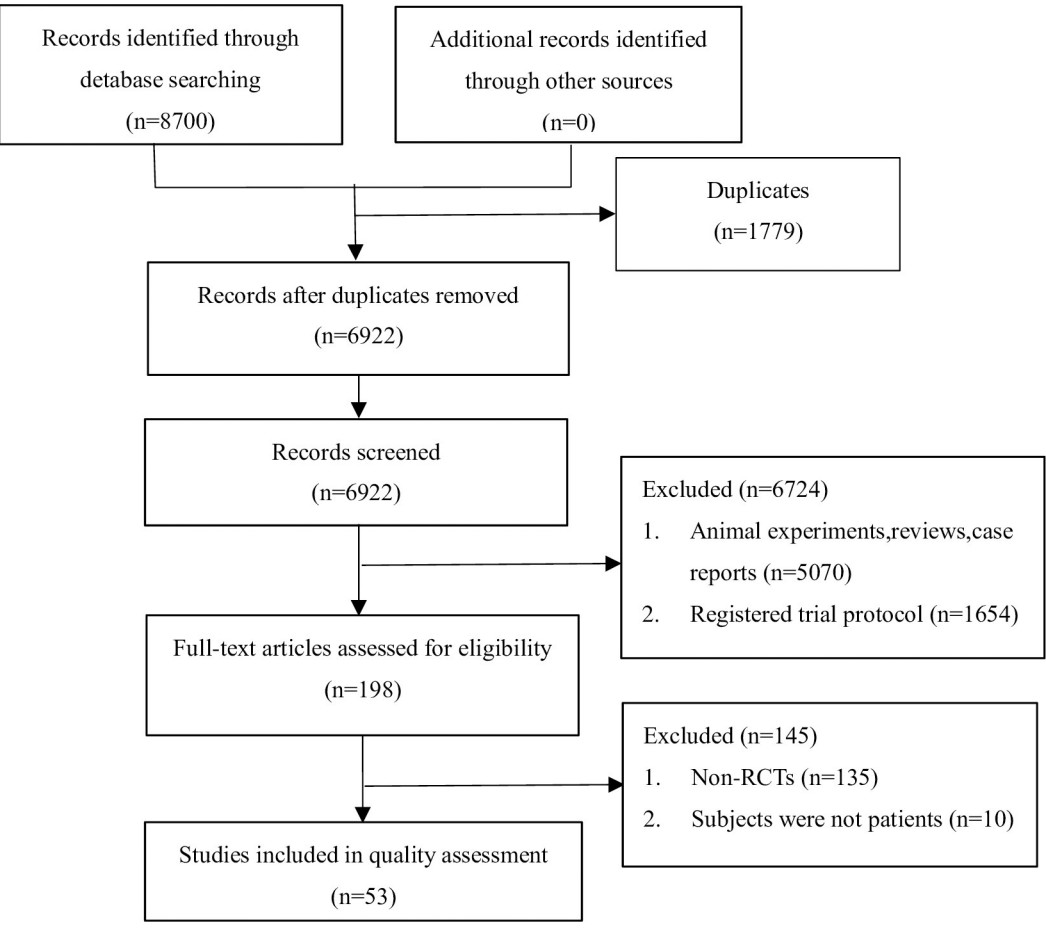

**Fig 1. Literature screening process and results.**

full text of each article was retrieved, and 53 RCTs were confirmed for further assessment. Fig 1 outlined the search detail via the PRISMA flow diagram.

## Agreement of reviewers

In the pilot study, inter-observer concordance for article selection had a kappa score of 0.78, which was 0.89 after resolving all disputed items by a discussion with the third reviewer (ZJX), suggesting that inter-observer reliability was almost perfect.

## Characteristics of included studies

Among the 53 RCTs, the first author of most RCTs were from Asia and accounted for 62.26%, half of the studies had a sample size greater than 100, 50 studies (94.34%) used drug interventions. 22 (42.51%) studies had an impact factor of more than 10. Nearly half of the studies were published in journals that did not explicitly require authors to follow the CONSORT statement. As showed in Table 1.

## The evaluation result of the CONSORT checklist

The study showed that the average reporting rate for 37 items in CONSORT checklist was 53.85%, with 31.35% for the methodological section. Items with the reporting rate of more than 80% were

**Table 1. Characteristics of included studies.**

| Characteristics | | n (%) | CONSORT score ($\bar{x} \pm$ SD) | F/t | P |
|---|---|---|---|---|---|
| Continent of first-author | | | | 0.205[a] | 0.892 |
| | Asia | 33 (62.26) | 13.09±3.884 | | |
| | Europe | 7 (13.21) | 12.79±2.325 | | |
| | North America | 8 (15.09) | 13.56±3.479 | | |
| | Others (South America and Africa) | 5 (9.43) | 12.00±3.391 | | |
| Number of authors | | | | 2.972[a] | 0.060 |
| | 1~10 | 9 (16.98) | 11.33±3.162 | | |
| | 11~20 | 13 (24.53) | 10.96±2.436 | | |
| | >20 | 31 (58.49) | 14.37±3.490 | | |
| Sample size | | | | -4.065[b] | <0.001 |
| | <100 | 25 (47.17) | 11.18±2.610 | | |
| | ≥100 | 28 (52.83) | 14.66±3.499 | | |
| Participants | | | | -0.304[b] | 0.763 |
| | suspected cases | 11 (20.75) | 12.73±2.563 | | |
| | confirmed cases | 42 (79.25) | 13.08±3.774 | | |
| Interventions | | | | -2.469[b] | 0.017 |
| | psychology | 3 (5.66) | 8.33±1.528 | | |
| | drugs | 50 (94.34) | 13.30±3.439 | | |
| Journal Impact Factor | | | | -6.101[b] | <0.001 |
| | <10 | 31 (58.49) | 11.10±2.361 | | |
| | ≥10 | 22 (42.51) | 15.73±3.169 | | |
| Endorsement of CONSORT | | | | -4.954[b] | <0.001 |
| | No | 23 (43.40) | 10.90±2.363 | | |
| | Yes | 30 (56.60) | 14.91±3.375 | | |

[a] equal to F value

[b] equal to t value.

abstract, background, eligibility criteria for participants, outcomes, statistical methods, participant flow chart, baseline data, limitations, generalisability and funding. Except for non-essential items, the remaining items with the reporting rate of less than 20% were trial design, sample size, allocation concealment, implementation, blinding, ancillary analyses and protocol. Only 7 studies (13.21%) reported the methods of masking concealment and 9 studies (16.98%) reported the details of the blinding. Table 2 outlined the reporting frequency of each checklist item.

The CONSORT average score for all study was 13.02±3.546, 95%CI (12.06–13.97).

## Factors of effecting overall reporting score

Univariate results showed that CONSORT score was associated with sample size (P<0.001), types of intervention (P = 0.017), journal impact factor (P<0.001), and endorsement of CONSORT statement (P<0.001) (Table 1).

Table 3 displayed the results of multiple linear regression analysis. The four predictors were entered into a multivariable model (constant = 7.779, $R^2$ = 0.533, adjusted $R^2$ = 0.494, P<0.001). Among these, journal impact factor (P = 0.006), and journal with endorsement of CONSORT statement (P = 0.014) still persisted as noticeable predictors of reporting quality.

Journal impact factors less than 10 and more than 10 had statistically significant differences in endorsement of the CONSORT statement (P<0.001) (Table 4). There was a statistically

**Table 2. The evaluation result of the CONSORT checklist.**

| Section /Topic | Item | Item No | Checklist item | Number of articles/n (%) |
|---|---|---|---|---|
| Title and abstract | | 1a | Identification as a randomized trial in the title | 41 (77.36) |
| | | 1b | Structured summary of trial design, methods, results, and conclusions | 45 (84.91) |
| Introduction | Background | 2a | Scientific background and explanation of rationale | 53 (100.00) |
| | objectives | 2b | Specific objectives or hypotheses | 39 (73.58) |
| Methods | Trial design | 3a | Description of trial design (such as parallel, factorial) including allocation ratio | 5 (9.43) |
| | | 3b | Important changes to methods after trial commencement (such as eligibility criteria), with reasons | 0 (0.00) |
| | Participants | 4a | Eligibility criteria for participants | 47 (88.68) |
| | | 4b | Settings and locations where the data were collected | 41 (77.36) |
| | Interventions | 5 | The interventions for each group with sufficient details to allow replication, including how and when they were actually administered | 53 (100.00) |
| | Outcomes | 6a | Completely defined pre-specified primary and secondary outcome measures, including how and when they were assessed | 50 (94.34) |
| | | 6b | Any changes to trial outcomes after the trial commenced, with reasons | 0 (0.00) |
| | Sample size | 7a | How sample size was determined | 5 (9.43) |
| | | 7b | When applicable, explanation of any interim analyses and stopping guidelines | 3 (5.66) |
| | Randomization: Sequence generation | 8a | Method used to generate the random allocation sequence | 26 (49.06) |
| | | 8b | Type of randomization; details of any restriction (such as blocking and block size) | 26 (49.06) |
| | Allocation concealment mechanism | 9 | Mechanism used to implement the random allocation sequence (such as sequentially numbered containers), describing any steps taken to conceal the sequence until interventions were 0 assigned | 7 (13.21) |
| | Implementation | 10 | Who generated the random allocation sequence, who enrolled participants, and who assigned participants to interventions | 0 (0.00) |
| | Blinding | 11a | If done, who was blinded after assignment to interventions (for example, participants, care providers, those assessing outcomes) and how | 9 (16.98) |
| | | 11b | If relevant, description of the similarity of interventions | 9 (16.98) |
| | Statistical methods | 12a | Statistical methods used to compare groups for primary and secondary outcomes | 53 (100.00) |
| | | 12b | Methods for additional analyses, such as subgroup analyses and adjusted analyses | 4 (7.55) |
| Results | Participant flow(a diagram is strongly recommended) | 13a | For each group, the numbers of participants who were randomly assigned, received intended treatment, and were analysed for the primary outcome | 43 (81.83) |
| | | 13b | For each group, losses and exclusions after randomization, together with reasons | 23 (43.40) |
| | Recruitment | 14a | Dates defining the periods of recruitment and follow-up | 32 (60.37) |
| | | 14b | Why the trial ended or was stopped | 3 (5.66) |
| | Baseline data | 15 | A table showing baseline demographic and clinical characteristics for each group | 53 (100.00) |
| | Numbers analysed | 16 | For each group, number of participants (denominator) included in each analysis and whether the analysis was by original assigned groups | 35 (66.04) |
| | Outcomes and estimation | 17a | For each primary and secondary outcome, results for each group, and the estimated effect size and its precision (such as 95% confidence interval) | 42 (79.25) |
| | | 17b | For binary outcomes, presentation of both absolute and relative effect sizes is recommended | 28 (52.83) |
| | Ancillary analyses | 18 | Results of any other analyses performed, including subgroup analyses and adjusted analyses, distinguishing pre-specified from exploratory | 4 (7.55) |
| | Harms | 19 | All important harms or unintended effects in each group | 41 (77.36) |
| Discussion | Limitations | 20 | Trial limitations, addressing sources of potential bias, imprecision, and if relevant, multiplicity of analyses | 51 (96.23) |
| | Generalisability | 21 | Generalisability (external validity, applicability) of the trial findings | 51 (96.23) |
| | Interpretation | 22 | Interpretation consistent with results, balancing benefits and harms, and considering other relevant evidence | 44 (83.01) |
| Other information | Registration | 23 | Registration number and name of trial registry | 38 (71.70) |
| | Protocol | 24 | Where the full trial protocol can be accessed, if available | 6 (11.32) |
| | Funding | 25 | Sources of funding and other support (such as supply of drugs), role of funders | 46 (86.79) |

Table 3. Multiple linear regression determinants of reporting quality of RCTs.

| Characteristics | Unstandardized Coefficients | | Standardized Coefficients | t | P | 95% CI | |
|---|---|---|---|---|---|---|---|
| | B | SE | Beta | | | Lower | Upper |
| Sample size | 1.058 | 0.875 | 0.150 | 1.209 | 0.233 | -0.702 | 2.817 |
| Types of interventions | -0.370 | 1.350 | -0.028 | -0.274 | 0.785 | -3.084 | 2.343 |
| Journal impact factor for 2020 | 0.058 | 0.020 | 0.414 | 2.854 | **0.006** | 0.017 | 0.099 |
| Endorsement of CONSORT | 2.190 | 0.859 | 0.309 | 2.549 | **0.014** | 0.463 | 3.917 |

constant = 9.324, $R^2$ = 0.533, adjusted $R^2$ = 0.494, P = 0.000.

95% CI, 95% confidence interval for B; B, Spearman–Brown coefficient; SE, standard error.

Bold values are those indicating statistical significance.

significant difference in reporting quality between journals that required submission of CONSORT checklist and those that did not (P = 0.031) (Table 5).

## Discussion

To our knowledge, this is the first study to evaluate the reporting quality of RCTs regarding patients with COVID-19 by the CONSORT statement [7], this will provide important information for clinical decision makers. Our study showed that the overall reporting quality of RCTs regarding patients with COVID-19 was suboptimal. It is clear that the reporting quality of these studies needs to be improved, particularly in terms of methodology section.

Although the CONSORT statement was established to ensure the completeness and accuracy of RCT reports, our study found that some authors still reported their data selectively in a biased way, with the average reporting rate of 53.85%, none of the RCTs provided complete information as required, similar results have been found in other studies [11, 16–18]. This could be due to the large number of patients with COVID-19 emerging in a short period of time, in order to present positive results of various treatment regimens to readers as soon as possible, researchers may have paid more attention to the results of study than report specifications.

The key items with lower reporting rates were mainly concentrated in trial design, sample size, allocation concealment, and blinding. The complete description of trial design can provide readers with accurate research ideas and enable readers to better evaluate the trial results [19], but our study showed that only 5 (9.43%) studies reported the type of trial design. The neglect of two most important items in the methodology section (allocation concealment and details of blinding) was particularly worrisome, as these items are important information to ensure the authenticity of results [20]. Only 7 (13.21%) studies described the methods of allocation concealment, 9 (16.98%) studies reported the details of blinding. We found that some

Table 4. Relationship between journal impact factor and endorsement of CONSORT.

| Jourrnal impact factor for 2020 | N (%) | Endorsement of CONSORT, n (%) | | Submission of CONSORT checklist, n (%) | |
|---|---|---|---|---|---|
| | | Yes | No | Yes | No |
| <10 | 31 (58.49) | 11 (20.75) | 20 (37.74) | 4 (7.55) | 7 (13.21) |
| ≥10 | 22 (41.51) | 19 (35.85) | 3 (5.66) | 16 (30.19) | 3 (5.66) |
| P | | <0.001[a] | | 0.015[b] | |

[a] p-Value for Pearson Chi-square test

[b] p-Value for Fisher's exact test

**Table 5. The impact of different journal submission requirements on report quality.**

| category | n | CONSORT score |
|---|---|---|
| No special requirements | 11 | 12.89±2.583 |
| Submission of CONSORT checklist | 19 | 15.76±3.327 |
| t | | -2.279 |
| P | | 0.031 |

authors tend to write "single" or "double" blind rather than specifying exactly who were unaware of treatment identities. The low reporting rates of these items may reflect the lack of relevant knowledge of researchers to some extent, because the report of sequence generation and concealment of allocation need to have certain knowledge of clinical research methodology [21]. There was evidence that trials with inadequate or unclear allocation concealment overestimated the treatment effects up to 7% [22], and a meta-epidemiological study of blinding showed that unblinded RCTs overestimated the outcome effect by 0.56 standard deviations [23]. In spite of this, some studies still reported these items poorly [11, 17, 24]. A previous study showed that only 12% and 21% of RCTs reported the details of allocation concealment and blinding in four high-impact general medical journals [25]. What's more, a study of acute herpes zoster showed that none of RCTs reported blinding and masking [16]. Therefore, it is urgent for researchers to pay attention to the design and implementation of allocation concealment and blinding, in order to minimize measurement bias and improve the reporting quality of RCTs. The estimation of sample size can avoid false negative results between the intervention and control groups [26]. Our study showed that only 5 studies (9.43%) reported the details of sample size, which was similar to the results in the field of vascular and endovascular surgery, herpes zoster and plastic surgery [9, 16, 27]. Lack of the reporting of sample size estimation can prevent readers to verify the validity of the trial results [27], researchers should attach great importance to the report of sample size estimation, so as to provide scientific evidence for future clinical studies.

Our study showed that reporting quality was associated with journal impact factor and endorsement of the CONSORT statement, similar results have been found in other areas [15, 27–29]. Studies have shown that 80% of RCTs published in journals that do not endorse CONSORT guideline had defective reporting specifications [30]. We found that journals with impact factor of more than 10 have a higher endorsement of the CONSORT guideline. However, even though some journals have endorsed the CONSORT guideline, the reporting quality of published RCTs was still suboptimal, this may be due to a problem with the entry point for paper submission. We found that some journals only encourage researchers to follow the CONSORT guideline, but do not require authors to submit a CONSORT checklist, which was more common in journals with lower impact factor, this may be because journals with lower impact factor have less stringent policies for accepting and publishing papers. In contrast, journals with higher impact factor have stricter requirements for submission of articles, requiring authors to upload CONSORT checklist when submitting RCT, which forces researchers to write RCTs according to standard reporting specifications. Our study showed that articles that uploaded the CONSORT checklist had higher reporting quality than those that did not have strict submission requirements, therefore, strict submission requirements are the premise to improve the reporting quality of RCTs.

Academic journals are the main media for carrying and publishing papers, the relevant provisions in the manuscript contract will directly affect the quality of published papers [28, 30]. Thus, we suggest that editors should carefully assess whether their journals' submission requirements are normative, journals should not only endorse the CONSORT guideline, but

more importantly require authors to upload the CONSORT checklist as key material for the initial screening when submitting their RCTs. Similarly, peer reviewers should check the completeness and accuracy of CONSORT checklist when reviewing RCTs, editorial boards should also increase their oversight of the entire process from submission to publication, articles of lower reporting quality will not be published. However, in addition to problems in the whole process of submission requirements, review and publication, another potential reason for the poor reporting quality of RCTs is that the CONSORT statement was not publicized enough, researchers lack the awareness of report guideline [31]. A survey of the authors of 101 studies found that only 3% acknowledged the importance of RCT reports and followed the CONSORT guideline when writing papers [31], this suggests that improving researchers' awareness of the CONSORT guideline is critical to improve the reporting quality of RCTs. Therefore, on the one hand, journals should vigorously promote the CONSORT guideline and can add relevant knowledge of RCT reports to their subscription feeds. On the other hand, research institutions should also increase training in these problems to improve the reporting quality of RCTs.

## Limitations of this study

There are some limitations in our study. Although the literature retrieval, screening and quality evaluation were carried out simultaneously by two researchers, there was still some subjectivity. In addition, we only included RCTs of patients with COVID-19 from 4 databases, it could not represent the overall reporting quality of RCTs of COVID-19.

## Conclusions

The primary objective of our study was to provide readers a broad overview of the reporting characteristics of RCTs regarding patients with COVID-19. The overall reporting quality of these RCTs was suboptimal, thereby diminishing their potential usefulness, and it can not provide valid evidence for clinical decision-making and systematic reviews. Better reporting quality was associated with higher journal impact factor and endorsement of the CONSORT statement. More journals should endorse the CONSORT statement, authors should strictly follow the relevant provisions of the CONSORT guideline when writing the paper. Future RCTs should particularly focus on improvement of detailed reporting in allocation concealment, blinding and estimation of sample size.

## Supporting information

**S1 File. Search strategy.**
(DOCX)

## Acknowledgments

We thank Dr. Bin Ma (Evidence-Based Medicine Center, Institute of Traditional Chinese and Western Medicine, School of Basic Medical Sciences, Lanzhou University, Lanzhou, Gansu, China) for providing assistance with editing the final manuscript.

## Author Contributions

**Conceptualization:** Yuhuan Yin.

**Data curation:** Yuhuan Yin, Fugui Shi, Yiyin Zhang, Xiaoli Zhang, Jianying Ye.

**Formal analysis:** Yuhuan Yin.

**Funding acquisition:** Fugui Shi.

**Methodology:** Yuhuan Yin, Fugui Shi, Juxia Zhang.

**Supervision:** Juxia Zhang.

**Validation:** Yuhuan Yin, Yiyin Zhang, Xiaoli Zhang, Jianying Ye.

**Writing – original draft:** Yuhuan Yin.

**Writing – review & editing:** Yuhuan Yin, Fugui Shi, Juxia Zhang.

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
