## [Decision Letter · Decision Letter 0]

16 Jul 2021

PONE-D-21-13616

Reporting quality of randomised controlled trials regarding patients or suspected patients with COVID-19

PLOS ONE

Dear Dr. Zhang,

I have now received the comments from the reviewers of your manuscript and after careful consideration, we feel that it has merit but does not fully meet PLOS ONE's publication criteria as it currently stands, and will require Major Revision. The reviewers raised complementary concerns, mainly focusing on inter-rater reliability, the validity of CONSORT scores provided for major items, and relying on the average score to provide descriptive summary of a result. This is in addition to language and clarity aspects of writing. Please review and address these and the other comments, revise your manuscript according to the PLOS ONE's instructions, and submit the revision for consideration. We will review your manuscript again after receiving your revision, should you decide to proceed, and re-evaluate as to whether the revised manuscript is suitable for publication.

I hope that the comments will provide useful in allowing you to revise the manuscript to reach the standard required by the journal.

We look forward to receiving your revised manuscript.

Kind regards,

Daoud Al-Badriyeh

Academic Editor

PLOS ONE

Journal Requirements

Reviewers' comments:

Reviewer's Responses to Questions

**Comments to the Author**

1. Is the manuscript technically sound, and do the data support the conclusions?

Reviewer #1: Yes

Reviewer #2: Partly

2. Has the statistical analysis been performed appropriately and rigorously? 

Reviewer #1: Yes

Reviewer #2: Yes

3. Have the authors made all data underlying the findings in their manuscript fully available?

Reviewer #1: Yes

Reviewer #2: Yes

4. Is the manuscript presented in an intelligible fashion and written in standard English?

Reviewer #1: No

Reviewer #2: Yes

5. Review Comments to the Author

Reviewer #1: While the data appears to be well done and I feel the topic is germaine, there are numerous issues with noun-verb agreement, there are a few cases where two sentences should be one, as well as multiple other grammatical issues. I highlighted a few of these in the attached document. However, they were too numerous to highlight them all.

Reviewer #2: Thank you for giving me the opportunity to comment on this interesting submission.

There are a number of major limitations that I would like to draw to your attention.

1)There is no mention of inter-rater reliability when judging the items. This is a major and recurring problem in studies that attempt to measure whether trials have adhered to reporting guidelines or not. There is a lot of subjective judgement involved in making a decision. For this to be robust and rigorous, the assessors should be trained and standardized.

2) For instance, some of the items are marked as 0% adherence to CONSORT. One of the items is with regards to participant exclusion and losses after randomization. I can't believe this can be true, because any trial that has the CONSORT flowchart will definitely have reported this information. Also, there are a couple of CONSORT items which ask to report any changes to methods or outcomes, if any. These were marked as 0%. However, if they trialists had no changes, they would not have reported it, so it seems to me somewhat inaccurate to mark these as 0%.

3) A lot of the summary descriptive statistics report 'average' scores etc. I don't necessarily think the mean or average is a suitable summary statistic here given the likelihood of markedly skewed data.

4) The discussion is much too long. I don't get a feel why this topic is important in context of other studies that have already demonstrated variable levels of adherence to CONSORT. I don't see that trials in COVID are any different from trials in cancer, surgery, diabetes, heart disease etc.

5) I think the analysis is unnecessarily complicated and there are too many variables being evaluated in a fishing expedition. I don't see that number of authors etc. has any bearing on CONSORT adherence for instance, and I can't understand why it was even entered into the statistical analysis.

6. PLOS authors have the option to publish the peer review history of their article (what does this mean?). If published, this will include your full peer review and any attached files.

Reviewer #1: No

Reviewer #2: **Yes: **YK Loke

---

## [Author Response · Author response to Decision Letter 0]

10 Aug 2021

Dear reviewer

We appreciate the positive comments which you made. Our responses to the comments are as follows:

Reviewer #1

While the data appears to be well done and I feel the topic is germaine, there are numerous issues with noun-verb agreement, there are a few cases where two sentences should be one, as well as multiple other grammatical issues. I highlighted a few of these in the attached document. However, they were too numerous to highlight them all.

Answer: We have reconstructed the grammar and writing in accordance with reviewer, please check.

Reviewer #2

1.There is no mention of inter-rater reliability when judging the items. This is a major and recurring problem in studies that attempt to measure whether trials have adhered to reporting guidelines or not. There is a lot of subjective judgement involved in making a decision. For this to be robust and rigorous, the assessors should be trained and standardized.

Answer: We have calculated inter-rater reliability in accordance with reviewer, the result was showed in page 8, please check.

2.For instance, some of the items are marked as 0% adherence to CONSORT. One of the items is with regards to participant exclusion and losses after randomization. I can't believe this can be true, because any trial that has the CONSORT flowchart will definitely have reported this information. Also, there are a couple of CONSORT items which ask to report any changes to methods or outcomes, if any. These were marked as 0%. However, if they trialists had no changes, they would not have reported it, so it seems to me somewhat inaccurate to mark these as 0%.

Answer: Because we initially had a wrong understanding of the item of participant exclusion and losses after randomization, we focused on the description of the results and ignored the information in the CONSORT flow chart. According to the reviewer’s opinion, we extracted the information of this item again (Table 2), and the results are shown in Table 2, please check.

 Other items reported as 0% are described in the methods section on page 6, items 3b, 6b, and 14b are not necessarily applicable, if relevant, the article will be graded according to the above guidelines, if the article did not apply to this item, no points was deducted, please check.

3.A lot of the summary descriptive statistics report 'average' scores etc. I don't necessarily think the mean or average is a suitable summary statistic here given the likelihood of markedly skewed data.

Answer: We have removed the summary statistical report of ‘average’ scores in accordance with reviewer, please check.

4.The discussion is much too long. I don't get a feel why this topic is important in context of other studies that have already demonstrated variable levels of adherence to CONSORT. I don't see that trials in COVID are any different from trials in cancer, surgery, diabetes, heart disease etc. I think the analysis is unnecessarily complicated and there are too many variables being evaluated in a fishing expedition. I don't see that number of authors etc. has any bearing on CONSORT adherence for instance, and I can't understand why it was even entered into the statistical analysis.

Answer: As no study has specifically evaluated the reporting quality of RCTs regarding patients with COVID-19. The primary objective of our study was to assess the reporting quality of RCTs regarding patients with COVID-19 and analyze possible related factors, so as to provide more references for high-quality research. We have pruned the discussion part according to the reviewer's opinion, please check. 

According to the comments of reviewers, we readjusted the variable analysis. We conducted univariate analysis in Table 1, factors that were statistically significant in univariate analysis were included in multiple regression models, the result was showed Table 3 (page 12), please check.

---

## [Editor Report · Decision Letter 1]

24 Aug 2021

Evaluation of reporting quality of randomized controlled trials in patients with COVID-19 using the CONSORT statement

PONE-D-21-13616R1

Dear Dr. Zhang,

We’re pleased to inform you that your manuscript has been judged scientifically suitable for publication and will be formally accepted for publication once it meets all outstanding technical requirements.

Kind regards,

Daoud Al-Badriyeh

Academic Editor

PLOS ONE
---

## [Editor Report · Acceptance letter]

16 Sep 2021

PONE-D-21-13616R1 

Evaluation of reporting quality of randomized controlled trials in patients with COVID-19 using the CONSORT statement 

Dear Dr. Zhang:

I'm pleased to inform you that your manuscript has been deemed suitable for publication in PLOS ONE. Congratulations! Your manuscript is now with our production department. 

Kind regards, 

on behalf of

Dr. Daoud Al-Badriyeh 

Academic Editor

PLOS ONE